# Heteronuclear Dirhodium-Gold Anionic Complexes: Polymeric Chains and Discrete Units

**DOI:** 10.3390/polym12091868

**Published:** 2020-08-19

**Authors:** Estefania Fernandez-Bartolome, Paula Cruz, Laura Abad Galán, Miguel Cortijo, Patricia Delgado-Martínez, Rodrigo González-Prieto, José L. Priego, Reyes Jiménez-Aparicio

**Affiliations:** 1Departamento de Química Inorgánica, Facultad de Ciencias Químicas, Universidad Complutense de Madrid, Ciudad Universitaria, E-28040 Madrid, Spain; estefania.fernandez@imdea.org (E.F.-B.); paula.cruz@urjc.es (P.C.); laura.abad-galan@ens-lyon.fr (L.A.G.); miguelcortijomontes@ucm.es (M.C.); bermejo@ucm.es (J.L.P.); 2Unidad de Difracción de Rayos X, Centro de Asistencia a la Investigación de Técnicas Físicas y Químicas, Universidad Complutense de Madrid, Ciudad Universitaria, E-28040 Madrid, Spain; patriciadelgado@ucm.es

**Keywords:** dirhodium(II) compounds, dicyano-aurate complexes, heteronuclear, one-dimensional, rhodium-gold anionic chains, coordination polymers

## Abstract

In this article, we report on the synthesis and characterization of the tetracarboxylatodirhodium(II) complexes [Rh_2_(μ–O_2_CCH_2_OMe)_4_(THF)_2_] (**1**) and [Rh_2_(μ–O_2_CC_6_H_4_–*p*–CMe_3_)_4_(OH_2_)_2_] (**2**) by metathesis reaction of [Rh_2_(μ–O_2_CMe)_4_] with the corresponding ligand acting also as the reaction solvent. The reaction of the corresponding tetracarboxylato precursor, [Rh_2_(μ–O_2_CR)_4_], with PPh_4_[Au(CN)_2_] at room temperature, yielded the one-dimensional polymers (PPh_4_)*_n_*[Rh_2_(μ–O_2_CR)_4_Au(CN)_2_]*_n_* (R = Me (**3**), CH_2_OMe (**4**), CH_2_OEt (**5**)) and the non-polymeric compounds (PPh_4_)_2_{Rh_2_(μ–O_2_CR)_4_[Au(CN)_2_]_2_} (R = CMe_3_ (**6**), C_6_H_4_–*p*–CMe_3_ (**7**)). The structural characterization of **1**, **3·2CH_2_Cl_2_**, **4·3CH_2_Cl_2_**, **5**, **6**, and **7·2OCMe_2_** is also provided with a detailed description of their crystal structures and intermolecular interactions. The polymeric compounds **3·2CH_2_Cl_2_**, **4·3CH_2_Cl_2_**, and **5** show wavy chains with Rh–Au–Rh and Rh–N–C angles in the ranges 177.18°–178.69° and 163.0°–170.4°, respectively. A comparative study with related rhodium-silver complexes previously reported indicates no significant influence of the gold or silver atoms in the solid-state arrangement of these kinds of complexes.

## 1. Introduction

Dirhodium(II) tetracarboxylato complexes with formula [Rh_2_(μ–O_2_CR)_4_] (R = alkyl or aryl) are an important part of the huge family of complexes with metal–metal bonds. They display a paddlewheel structure and a single metal–metal bond order due to their background electronic configuration is σ^2^π^4^δ^2^δ*^2^π*^4^, which is responsible for their diamagnetic nature [1,2,3]. The properties and reactivity of these complexes and their derivatives make them very interesting compounds for the scientific community. Due to their potential applications, they have been studied in fields like catalysis [4,5,6,7,8,9,10,11,12], bioinorganic chemistry [13,14,15,16,17], metal organic frameworks (MOFs) [18,19], or gas absorption [20,21]. Metal-organic aerogels [22,23] and liquid crystals [24,25] can also be obtained using them as building blocks.

The structural diversity found in many of this kind of complexes must be also highlighted [12,20,21,26,27,28,29,30,31,32,33]. The axial sites of the paddlewheel structure are easily occupied by monodentate ligands, which has allowed the preparation of a large number of molecular compounds [1,2,34,35,36,37]. One-dimensional polymers [1,2,38,39,40,41] can be obtained by means of bridging ligands between the dirhodium(II) cores.

Several approaches can be found in the literature to obtain heterometallic one-dimensional polymers where different metal complexes connect the dirhodium units [42,43,44,45,46,47,48,49,50]. The valuable physicochemical properties of some of them, like paramagnetism [42,47], modulation of their electronic structures [43], or luminescence [49], turn this kind of polymers into very promising materials. However, the number of heterometallic coordination polymers based on rhodium(II) carboxylates is still scarce.

Moreover, there is also an extensive bibliography about the use of cyanidometallate complexes to construct heteronuclear coordination polymers. This interest is explained due to their structural variety and interesting properties such as magnetism or luminescence [51,52,53,54,55,56,57,58]. Following this strategy, our research group has reported the synthesis and characterization of several heterometallic complexes based on cyanidometallate ligands coordinated to the axial positions of the Rh_2_^4+^ paddlewheel unit [59,60,61].

The reaction of the corresponding paddlewheel tetracarboxylatodirhodium with cyanidometallate complexes in solution at room temperature allowed the synthesis of polymeric chains with formula K*_n_*{Rh_2_(μ–O_2_CR)_4_[Au(CN)_2_]}*_n_* (R = Me, Et) [59], and, very recently, (PPh_4_)*_n_*[Rh_2_(μ–O_2_CR)_4_Ag(CN)_2_]*_n_* (R = Me, Ph, CH_2_OEt) [60] and (PPh_4_)_2*n*_[{Rh_2_(µ–O_2_CMe)_4_}{M(CN)_4_}]*_n_* (M = Ni, Pd, Pt) [61]. A similar reaction led also to the formation of the non-polymeric complex (PPh_4_)_2_{Rh_2_(μ–O_2_CCMe_3_)_4_[Ag(CN)_2_]_2_} [60]. The presence of [Au(CN)_2_]^−^ in these kind of complexes opens the possibility of aurophilic interactions [62] as it is found in compound K*_n_*{Rh_2_(μ–O_2_CEt)_4_[Au(CN)_2_]}*_n_* which displays luminescence with a broad intense emission at 475 nm upon excitation at 360 nm [59]. However, in spite of their easy synthesis and their potential luminescent properties, the complexes K*_n_*{Rh_2_(μ–O_2_CR)_4_[Au(CN)_2_]}*_n_* (R = Me, Et) [59] are, to our knowledge, the only two examples of polymers based on dirhodium tetracarboxylates with dicyanidoaurate(I) as axial bridge. Moreover, differences in the supramolecular structures of these complexes cause also differences in the luminescent properties, as the methyl derivative do not show this feature due to its long Au⋯Au distances. This fact highlights the importance of increasing the number of this type of polymers that allow the study of the influence of the solid state arrangement in their properties. Additionally, the counterion plays also an important role on the crystal structure and possible supramolecular interactions. For example, the bulky tetraphenylphosphonium cation can form supramolecular architectures by means of phenyl–phenyl embraces [63,64].

Taking into account the antecedents mentioned above, in this article we report the synthesis, characterization, and structural description of three heterometallic dirhodium-gold polymeric complexes with the formula (PPh_4_)*_n_*[Rh_2_(μ–O_2_CR)_4_Au(CN)_2_]*_n_* (R = Me (**3**), CH_2_OMe (**4**), CH_2_OEt (**5**)). The structure of the starting complex [Rh_2_(μ–O_2_CCH_2_OMe)_4_(THF)_2_] (**1**) is also described. The same reactions conditions starting from [Rh_2_(μ–O_2_CCMe_3_)_4_(HO_2_CCMe_3_)_2_] and [Rh_2_(μ–O_2_CC_6_H_4_–*p*–CMe_3_)_4_(OH_2_)_2_] (**2**) led to the non-polymeric complexes (PPh_4_)_2_{Rh_2_(μ–O_2_CR)_4_[Au(CN)_2_]_2_} (R = CMe_3_ (**6**), C_6_H_4_–*p*–CMe_3_ (**7**)), respectively. The structural characterization of compounds **6** and **7** is also provided in this work. The comparison of complexes **3**, **5**, and **6** with their silver derivatives [60] allows the study of the influence of gold or silver atoms in the crystal structures. Intermolecular interactions have been carefully surveyed in order to find possible phenyl embraces between the phenyl rings or short Au⋯Au distances.

## 2. Materials and Methods

### 2.1. Materials

[Rh_2_(μ–O_2_CCH_2_OEt)_4_(HO_2_CCH_2_OEt)_2_] [60] and [Rh_2_(μ–O_2_CCMe_3_)_4_(HO_2_CCMe_3_)_2_] [65,66] were prepared following published procedures. PPh_4_[Au(CN)_2_] was synthesized following the published method to obtain PPh_4_[Ag(CN)_2_] [60]. A solution of 0.2 mmol (0.06 g) of K[Au(CN)_2_] in 4 mL of water was mixed with a solution of 0.2 mmol (0.08 g) of PPh_4_Br in 8 mL of water and stirred for 5 min at room temperature. The white precipitate obtained was collected by filtration and washed with 15 mL of water and 10 mL of diethyl ether (10 mL). Yield: 0.082 g (70%). The rest of the reagents and solvents were acquired from commercial sources and used as received without further purification.

### 2.2. Physical Measurements

The elemental analysis measurements were carried out at the Microanalytical Services of the Complutense University of Madrid. FTIR measurements were carried out in the 4000 to 650 cm^−1^ spectral range with a Perkin–Elmer Spectrum 100 equipped with an universal ATR accessory (PerkinElmer, Inc., Shelton, CT, USA).

### 2.3. Crystallography

Single-crystal X-ray diffraction measurements were carried out at room temperature using a Bruker Smart-CCD diffractometer (Bruker Corporation, Billerica, MA, USA) with a Mo Kα (λ = 0.71073 Å) radiation and a graphite monochromator. CCDC 2015497–2015501 and 2015886 contain the crystallographic data for the compounds described in this article. These data can be obtained free of charge from the Cambridge Crystallographic Data Centre via *www.ccdc.cam.ac.uk/data_request/cif*.

### 2.4. Synthesis

#### 2.4.1. Synthesis of [Rh_2_(μ–O_2_CCH_2_OMe)_4_(THF)_2_] (**1**)

A mixture of 0.68 mmol (0.30 g) of [Rh_2_(μ–O_2_CMe)_4_] and 29.85 mmol (2.69 g, 2.29 mL) of methoxyacetic acid was stirred and heated at 120 °C for 30 min under nitrogen atmosphere. The mixture was allowed to cool and the sticky product obtained was triturated and washed with 2 × 25 mL of a 3:2 hexane/diethyl ether mixture to obtain a green solid. Single crystals of **1** were obtained after 3 days by slow diffusion of hexane into a solution of the solid in THF. Yield: 0.25 g (52%). Anal. Calcd. (%) for [Rh_2_(μ–O_2_CCH_2_OMe)_4_]: C, 25.64; H, 3.59. Found (%): C, 25.88; H, 3.53. FT-IR (cm^−1^): 2935w, 2829w, 1600vs, 1431m, 1407s, 1330s, 1278w, 1197m, 1158w, 1130m, 1093s, 1016w, 939m, 898m, 731s.

#### 2.4.2. Synthesis of [Rh_2_(μ–O_2_CC_6_H_4_–*p*–CMe_3_)_4_(OH_2_)_2_] (**2**)

[Rh_2_(μ–O_2_CMe)_4_] (0.09 mmol (0.04 g)) and 40.00 mmol (7.13 g) of 4–*tert*–butylbenzoic acid were mixed and heated under nitrogen atmosphere until the latter melted (~165 °C). The reaction was kept for 30 min and then let to cool down to room temperature. The turquoise solid obtained was collected from the bottom of the flask and washed with several 60 mL fractions of a 1:5 diethyl ether/petroleum ether mixture. Yield: 0.02 g (23%). Anal. Calcd. (%) for **2**: C, 55.59; H, 5.94. Found (%): C, 55.52; H, 5.74. FT-IR (cm^−1^): 3416w, 2963m, 2906w, 2869w, 1607m, 1589m, 1546m, 1466w, 1391s, 1268m, 1192m, 1148w, 1109w, 1016m, 856m, 781m, 731m, 712m.

#### 2.4.3. Synthesis of (PPh_4_)*_n_*[Rh_2_(μ–O_2_CMe)_4_Au(CN)_2_]*_n_* (**3**)

A 7 mL THF solution of 0.18 mmol (0.08 g) of [Rh_2_(μ–O_2_CMe)_4_] was mixed with a 12 mL THF solution of 0.19 mmol (0.11 g) of PPh_4_[Au(CN)_2_] and stirred for 1 day at room temperature obtaining a purple precipitate. The solid was filtered and washed with THF. Yield: 0.08 g (43%). Anal. Calcd. (%) for **3**: C, 39.63; H, 3.13; N, 2.72. Found (%): C, 39.78; H, 3.16; N, 2.79. FT-IR (cm^−1^): 3083w, 2173w, 1598s, 1484w, 1409s, 1342m, 1163w, 1109s, 1042m, 997m, 753m, 721s, 688s.

The solid was dissolved in dichloromethane and THF was slowly added on top of the solution. Violet single crystals of **3·2CH_2_Cl_2_** were obtained after 2 days.

#### 2.4.4. Synthesis of (PPh_4_)*_n_*[Rh_2_(μ–O_2_CCH_2_OMe)_4_Au(CN)_2_]*_n_* (**4**)

The synthesis was similar to the synthesis of **3** although in this case a solution of 0.11mmol (0.08 g) of [Rh_2_(μ–O_2_CCH_2_OMe)_4_(THF)_2_] (**1**) in 5 mL of methanol and a solution of 0.12 mmol (0.07 g) of PPh_4_[Au(CN)_2_] in 8 mL of THF were employed. The mixture was stirred for 30 min obtaining a purple solution. The solvent was evaporated and the solid obtained was washed with cold THF. Yield: 0.05 g (40%). Anal. Calcd. (%) for **4**: C, 39.67; H, 3.50; N, 2.43. Found (%): C, 40.03; H, 3.62; N, 2.31. FT-IR (cm^−1^): 2822w, 2139w, 1608s, 1484w, 1435m, 1412m, 1330m, 1189w, 1162w, 1109vs, 1027w, 996w, 924w, 849w, 752w, 732vs, 687s.

Single crystals of **4·3CH_2_Cl_2_** suitable for X-ray diffraction were obtained after 4 days by slow diffusion of hexane into a solution of the compound in 4 mL of dichloromethane.

#### 2.4.5. Synthesis of (PPh_4_)*_n_*[Rh_2_(μ–O_2_CCH_2_OEt)_4_Au(CN)_2_]*_n_* (**5**)

The synthesis was analogous to that of **3** using 8 mL of a THF solution of 0.07 mmol (0.06 g) of [Rh_2_(μ–O_2_CCH_2_OEt)_4_(HO_2_CCH_2_OEt)_2_] and 0.10 mmol (0.06 g) of PPh_4_[Au(CN)_2_] in 12 mL of THF. The purple solid obtained was washed with THF. Yield: 0.07 g (83%). Anal. Calcd. (%) for **5**: C, 41.81; H, 4.01; N, 2.32. Found (%): C, 41.58; H, 3.98; N, 2.39. FT-IR (cm^−1^): 3074w, 2966m, 2925m, 2161m, 1612s, 1485m, 1433s, 1407s, 1363m, 1320s, 1260m, 1163m, 1135s, 1107s, 1032m, 1008m, 998m, 894w, 851m, 756m, 724s, 694s.

Purple single crystals of **5** were obtained after 2 days by slow diffusion of diethyl ether in a dichloromethane solution of the compound.

#### 2.4.6. Synthesis of (PPh_4_)_2_{Rh_2_(μ–O_2_CCMe_3_)_4_[Au(CN)_2_]_2_} (**6**)

The synthesis was analogous to that of **3** using a solution of 0.10 mmol (0.08 g) of [Rh_2_(μ–O_2_CCMe_3_)_4_(HO_2_CCMe_3_)_2_] in 10 mL of diethyl ether and a solution of 0.10 mmol (0.06 g) of PPh_4_[Au(CN)_2_] in 8 mL of acetone. Yield: 0.034 g (38%). Anal. Calcd. (%) for **6**: C, 48.39; H, 4.29; N, 3.14. Found (%): C, 47.58; H, 4.20; N, 3.16. FT-IR (cm^−1^): 3061w, 2968w, 2932w, 2866w, 2148w, 1576s, 1482s, 1458m, 1439s, 1412s, 1373m, 1361m, 1220s, 1107s, 997m, 935s, 894w, 802w, 781w, 763m, 723s, 690s.

Purple single crystals of **6** were obtained by slow evaporation of a solution of the solid in a 1:1 acetone/diethyl ether mixture.

#### 2.4.7. Synthesis of (PPh_4_)_2_{Rh_2_(μ–O_2_CC_6_H_4_–*p*–CMe_3_)_4_[Au(CN)_2_]_2_} (**7**)

The synthesis was analogous to that of **3** but using dichloromethane solutions of the reactants 0.10 mmol (0.09 g) of [Rh_2_(μ–O_2_CC_6_H_4_–*p*–CMe_3_)_4_(OH_2_)_2_] (**2**) and 0.10 mmol (0.06 g) of PPh_4_[Au(CN)_2_]. A solution was obtained after the reaction, the solvent was evaporated, and a purple solid was obtained and washed with a dichloromethane/petroleum ether mixture. Yield: 0.07 g (67%). Anal. Calcd. (%) for **7·**2CH_2_Cl_2_: C, 52.05; H, 4.28; N, 2.25. Found (%): C, 52.58; H, 4.75; N, 2.48. FT-IR (cm^−1^): 3058w, 2961m, 2906w, 2866w, 2161w, 1595s, 1556m, 1484w, 1474w, 1462w, 1437m, 1393s, 1268m, 1190m, 1149w, 1107s, 1017m, 997w, 855m, 780m, 755w, 722s, 712s, 689s.

Single crystals of (PPh_4_)_2_{Rh_2_(μ–O_2_CC_6_H_4_–*p*–CMe_3_)_4_[Au(CN)_2_]_2_}·2OCMe_2_ (**7·2OCMe_2_**) were obtained after 7 days by slow diffusion of hexane into a solution of the compound in acetone in the fridge.

## 3. Results and discussion

### 3.1. Synthesis of the Complexes

Complexes **1**–**7** have been obtained following the routes indicated in Scheme 1.

The metathesis reaction of [Rh_2_(μ–O_2_CMe)_4_] in methoxyacetic acid led to the formation of [Rh_2_(μ–O_2_CCH_2_OMe)_4_(THF)_2_] (**1**) after the removal of the excess ligand, washing with hexane and diethyl ether and crystallization process in THF. A similar reaction with melted 4–*tert*–butylbenzoic acid yielded complex [Rh_2_(μ–O_2_CC_6_H_4_–*p*–CMe_3_)_4_(OH_2_)_2_] (**2**) after washing with a mixture of diethyl ether and petroleum ether to eliminate the huge excess of solid ligand. The low solubility of the 4–*tert*–butylbenzoic acid in the washing mixture made necessary the use of a large volume of solvent. This synthetic method using the ligand as the reaction solvent is similar to the one used for the synthesis of [Rh_2_(μ–O_2_CCH_2_OEt)_4_(HO_2_CCH_2_OEt)_2_] [60] and [Rh_2_(μ–O_2_CCMe_3_)_4_(HO_2_CCMe_3_)_2_] [65,66]. Slow diffusion of different solvents into solutions of the complexes **1** and **2** was tested in order to get single crystals to allow the structural determination of these compounds. Single crystals of **1** were obtained using hexane and a solution of the solid obtained in THF. Unfortunately, all the attempts to obtain single crystals of **2** were unsuccessful. Nevertheless, the paddlewheel structure of complex **2** with four 4–*tert*–butylbenzoate bidentate ligands is proved in the crystal structure of complex **7** (see below).

One-dimensional polymeric complexes, (PPh_4_)*_n_*[Rh_2_(μ–O_2_CMe)_4_Au(CN)_2_]*_n_* (**3**), (PPh_4_)*_n_*[Rh_2_(μ–O_2_CCH_2_OMe)_4_Au(CN)_2_]*_n_* (**4**), and (PPh_4_)*_n_*[Rh_2_(μ–O_2_CCH_2_OEt)_4_Au(CN)_2_]*_n_* (**5**), were obtained by stirring at room temperature solutions of the corresponding tetracarboxylato dirhodium complex with a solution of PPh_4_[Au(CN)_2_] in THF. For compounds **3** and **5**, THF solutions were employed and these compounds were obtained as precipitates from the reaction mixture. For compound **4**, the chosen solvent was methanol and the solid was obtained after removal of the solvent mixture. Single crystals of the three complexes were obtained from slow diffusion of different solvents into dichloromethane solutions of the solids. In this way, THF was used to obtain single crystals of {(PPh_4_)[Rh_2_(μ–O_2_CMe)_4_Au(CN)_2_]·2CH_2_Cl_2_}*_n_* (**3·2CH_2_Cl_2_**), hexane for {(PPh_4_)[Rh_2_(μ–O_2_CCH_2_OMe)_4_Au(CN)_2_]·3CH_2_Cl_2_]}*_n_* (**4·3CH_2_Cl_2_**), and diethyl ether for (PPh_4_)*_n_*[Rh_2_(μ–O_2_CCH_2_OEt)_4_Au(CN)_2_]*_n_* (**5**). Single crystals of compound **5**, with the same unit cell, were also obtained by slow diffusion of diethyl ether in an acetone solution of the compound.

Similar reactions at room temperature were employed to obtain the non-polymeric complexes (PPh_4_)_2_{Rh_2_(μ–O_2_CCMe_3_)_4_[Au(CN)_2_]_2_} (**6**) and (PPh_4_)_2_{Rh_2_(μ–O_2_CC_6_H_4_–*p*–CMe_3_)_4_[Au(CN)_2_]_2_} (**7**). An acetone solution of PPh_4_[Au(CN)_2_] was stirred with a solution of [Rh_2_(μ–O_2_CCMe_3_)_4_(HO_2_CCMe_3_)_2_] in diethyl ether to obtain compound **6**. Dichloromethane solutions of PPh_4_[Au(CN)_2_] and **2** achieved compound **7** after evaporation of the solvent. Single crystals of **6** were formed after evaporation of a solution of the complex in acetone/diethyl ether, whereas layering hexane on top of an acetone solution of **7** and kept in the fridge (7 days) yielded single crystals of (PPh_4_)_2_{Rh_2_(μ–O_2_CC_6_H_4_–*p*–CMe_3_)_4_[Au(CN)_2_]_2_}·2OCMe_2_ (**7·2OCMe_2_**).

The reaction conditions used for the synthesis of compounds **3**–**7** are analogous to those employed for the synthesis of the polymeric complexes (PPh_4_)*_n_*[Rh_2_(μ–O_2_CR)_4_Ag(CN)_2_]*_n_* (R = Me, Ph, CH_2_OEt) and the non-polymeric compound, (PPh_4_)_2_{Rh_2_(μ–O_2_CCMe_3_)_4_[Ag(CN)_2_]_2_} recently reported by our research group [60]. This lead to crystals structures showing numerous structural similarities (see below).

### 3.2. IR Characterization

The IR spectrum of each compound shows the characteristic bands of the carboxylate ligands coordinated to the dimetallic unit. Thus, the infrared spectra of the seven compounds show the corresponding bands of the antisymmetric and symmetric stretching modes of the carboxylate groups: ν(COO)_a_ (1612–1576 cm^−1^) and ν(COO)_s_ (1412–1391 cm^−1^). The separation between the symmetric and antisymmetric bands indicates the symmetrical bridging coordination mode of the equatorial carboxylate ligands [67].

For complexes **3**–**7**, an additional band corresponding to the ν(C≡N) vibration is also visible in the 2173 to 2139 cm^−1^ range, indicating the presence of the [Au(CN_)2_] in the complexes.

### 3.3. Crystal Structures and Refinement Data

A summary of some crystal and refinement data obtained for complexes **1**, **3·2CH_2_Cl_2_**, **4·3CH_2_Cl_2_**, **5**, **6**, and **7·2OCMe_2_** is shown in Table 1. More detailed information can be found in Appendix A

[Rh_2_(μ–O_2_CCH_2_OMe)_4_(THF)_2_] (**1**), {(PPh_4_)[Rh_2_(μ–O_2_CMe)_4_Au(CN)_2_]·2CH_2_Cl_2_}*_n_* (**3·2CH_2_Cl_2_**), {(PPh_4_)[Rh_2_(μ–O_2_CCH_2_OMe)_4_Au(CN)_2_]·3CH_2_Cl_2_]}*_n_* (**4·3CH_2_Cl_2_**), (PPh_4_)_2_{Rh_2_(μ–O_2_CCMe_3_)_4_[Au(CN)_2_]_2_} (**6**), and (PPh_4_)_2_{Rh_2_(μ–O_2_CC_6_H_4_–*p*–CMe_3_)_4_[Au(CN)_2_]_2_}·2OCMe_2_ (**7·2OCMe_2_**) crystallize in the triclinic *P**-1* space group, while (PPh_4_)*_n_*[Rh_2_(μ–O_2_CCH_2_OEt)_4_Au(CN)_2_]*_n_* (**5**) crystallizes in the monoclinic *P*2_1_/c space group. Compounds **3·2CH_2_Cl_2_** and **5** crystallize in the same space groups with similar cell parameters than the related dicyanidoargentate(I) complexes {(PPh_4_)[Rh_2_(μ–O_2_CMe)_4_Ag(CN)_2_]·2CH_2_Cl_2_}*_n_* and (PPh_4_)*_n_*[Rh_2_(μ–O_2_CCH_2_OEt)_4_Ag(CN)_2_]*_n_* [60]. However, **6** crystallizes in a different space group than (PPh_4_)_2_{Rh_2_(μ–O_2_CCMe_3_)_4_[Ag(CN)_2_]_2_}, which crystallizes in the orthorhombic *Pnma* space group [60].

The asymmetric unit of **1** contains two halves of two different paddlewheel units with inversion centers placed in between the M–M axis (Appendix A). Similarly, the asymmetric unit of both complexes **6** and **7·2OCMe_2_** contains only one half of a paddlewheel unit and the other half is generated in the unit cell by an inversion center located in the center of the M–M axis. A tetraphenylphosphonium cation in both structures and one acetone molecule in **7·2OCMe_2_** complete the asymmetric unit of these compounds (Appendix A).

The asymmetric unit of **3·2CH_2_Cl_2_** and **4·3CH_2_Cl_2_** and **5** is formed by a tetraphenylphosphonium cation and a negatively charged tetracarboxylatodirhodium(II) unit with a dicyanidoaurate(I) ligand in one axial position. Additionally, dichloromethane molecules complete the asymmetric unit of **3·2CH_2_Cl_2_** and **4·3CH_2_Cl_2_** (see Appendix A). The quality of the data allowed to model isotropically two and three dichloromethane molecules in **3·2CH_2_Cl_2_** and **4·3CH_2_Cl_2_**, respectively. The Platon Squeeze Routine [68] was used to remove additional disordered electron density in the structure of **3·2CH_2_Cl_2_**.

All the Rh(II) ions display a slightly distorted octahedral geometry, with the four equatorial positions occupied by oxygen atoms of the carboxylate ligands and the axial positions occupied by another Rh(II) ion, and the oxygen atom of a THF molecule in the case of **1**, and a nitrogen atom of a dicyanidoaurate(I) ligand in the case of **3·2CH_2_Cl_2_**, **4·3CH_2_Cl_2_**, **5**, **6**, and **7·2OCMe_2_**. Rh-O_equatorial_ distances are in the 1.989(1) to 2.063(5) Å range. Octahedra are axially elongated with Rh-O_axial_ distances of 2.256(3) and 2.258(3) Å for **1**, Rh-N_axial_ distances of 2.221(7) and 2.223(7) Å for **3·2CH_2_Cl_2_**, 2.209(8) and 2.187(9) Å for **4·3CH_2_Cl_2_**, 2.249(6) and 2.238(5) Å for **5**, 2.226(4) Å for **6** and 2.291(11) Å for **7·2OCMe_2_**, and Rh-Rh distances of 2.3787(8) and 2.3810(8) Å for **1**, 2.3981(9) Å for **3·2CH_2_Cl_2_**, 2.4096(11) Å for **4·3CH_2_Cl_2_**, 2.4133(8) Å for **5**, 2.4002(6) Å for **6** and 2.3969(19) Å for **7·2OCMe_2_**. All the Rh-Rh distances are indicative of a single bond between Rh(II) ions. Rh-Rh and Rh-N distances are similar to those found in other related compounds such as (PPh_4_)_2*n*_[{Rh_2_(µ–O_2_CCH_3_)_4_}{M(CN)_4_}]*_n_* (M = Ni, Pd, Pt) [61], K*_n_*[Rh_2_(µ–O_2_CR)_4_Au(CN)_2_]*_n_* (R = Me, Et) [59], (PPh_4_)*_n_*[Rh_2_(µ–O_2_CR)_4_Ag(CN)_2_]*_n_* (R = Me, Ph, CH_2_OEt) [60], (PPh_4_)_2_{Rh_2_(μ–O_2_CCMe_3_)_4_[Ag(CN)_2_]_2_} [60], [K(18–crown–6)(H_2_O)]_2*n*_[K(18–crown–6)(H_2_O)_2_]*_n_*[Rh_2_(µ–O_2_CPh)_4_Fe(CN)_6_]*_n_*·8nH_2_O [69], and K_3*n*_{[Rh_2_(µ–O_2_CCH_3_)_4_]_2_Co(CN)_6_}*_n_* [70].

The structure of **3·2CH_2_Cl_2_, 4·3CH_2_Cl_2_**, and **5** is formed by chains constructed, respectively, by the following repetitive anionic units; [Rh_2_(μ–O_2_CMe)_4_Au(CN)_2_]^−^, [Rh_2_(μ–O_2_CCH_2_OMe)_4_Au(CN)_2_]^−^, and [Rh_2_(μ-O_2_CCH_2_OEt)_4_Au(CN)_2_]^−^. A representation of the polymeric structure observed in **3·2CH_2_Cl_2_** is shown in Figure 1a as an example. On the other hand, the structure of **1**, **6**, and **7·2OCMe_2_** is formed by discrete [Rh_2_(μ–O_2_CCH_2_OMe)_4_(THF)_2_], {Rh_2_(μ–O_2_CCMe_3_)_4_[Au(CN)_2_]_2_}^2−^, and {Rh_2_(μ–O_2_CC_6_H_4_–*p*–CMe_3_)_4_[Au(CN)_2_]_2_}^2−^ units, respectively. The anionic paddlewheel unit of the structure of **6** is shown in Figure 1b. The crystal structure of complex **6** is similar to its analogous compound (PPh_4_)_2_{Rh_2_(μ–O_2_CCMe_3_)_4_[Ag(CN)_2_]_2_}, whose molecular nature has been explained by a higher solubility of the branched trimethylacetate group in acetone, favoring the formation of a molecular complex due to a slower crystallization [60]. This fact is also corroborated by the obtaining of discrete anionic {Rh_2_(μ–O_2_CC_6_H_4_–*p*–CMe_3_)_4_[Au(CN)_2_]_2_}^2−^ units in complex **7**, due to the also branched 4–*tert*–butylbenzoate ligand. A view of the packing of the discrete dirhodium units of **6** and **7·2OCMe_2_** is shown in Appendix A Information.

The anionic chains of **3·2CH_2_Cl_2_**, **4·3CH_2_Cl_2_**, and **5** have a wavy structure with Rh-Au-Rh angles of 178.69(1), 177.99(2), and 177.18(1), respectively, and Rh–N–C angles of 169.8(8) and 170.2(8) for **3·2CH_2_Cl_2_**, 170.4(10) and 167.7(10) for **4·3CH_2_Cl_2_** and 164.2(7) and 163.0(6) for **5**. This wavy structure is analogous to those found in the related compounds (PPh_4_)*_n_*[Rh_2_(µ–O_2_CR)_4_Ag(CN)_2_]*_n_* (R = Me, Ph, CH_2_OEt) [60] with very similar values for the Rh-M-Rh (M = Ag, Au) and Rh–N–C angles when complexes with the same equatorial ligand are compared. These angles make that the wavy structures for compounds with R = CH_2_OEt are more pronounced that for those with R = Me.

The anionic chains of **3·2CH_2_Cl_2_** and **4·3CH_2_Cl_2_** are packed parallel to each other along the *a* axis with the tetraphenylphosphonium cations and dichloromethane molecules between them. The chains of **3·2CH_2_Cl_2_** are arranged in pairs with each pair surrounded by other four pairs of chains (Figure 2a) in a similar way than the structure found for {(PPh_4_)[Rh_2_(μ–O_2_CMe)_4_Ag(CN)_2_]·2CH_2_Cl_2_}*_n_* [60], while the chains are arranged in rows in the structure of **4·3CH_2_Cl_2_** (Figure 2b). The anionic chains of **5** are packed in a parallel disposition, with tetraphenylphosphonium cations between them, in an alternating fashion as shown in Figure 2c. This chain packing is comparable to that found in the silver derivative, (PPh_4_)*_n_*[Rh_2_(μ–O_2_CCH_2_OEt)_4_Ag(CN)_2_]*_n_* [60].

The supramolecular interactions between dirhodium units of the compounds are summarized in this paragraph. Several CH⋯O contacts between dirhodium molecules are observed in the structure of **1**. Each Rh1Rh1 unit is connected to four neighbor molecules and each Rh2Rh2 unit is connected to six neighbor molecules through this type of contacts (Appendix A). Two CH⋯O contacts link couples of neighbor chains in the structure of **4·3CH_2_Cl_2_** (Appendix A), and two CH⋯N contacts connect the dirhodium discrete units in one direction in the structure of **7·2OCMe_2_** (Appendix A). There is no significant interaction between dirhodium units in the structure of **3·2CH_2_Cl_2_**, **5**, and **6**. Moreover, the presence of tetraphenylphosphonium cations surrounding the dirhodium units does not allow the existence of Au–Au interactions in any structure. Thus, the shortest Au⋯Au distances are 8.7190(8), 9.747(2), 8.452(2), 7.8186(7), and 7.155(2) Å for **3·2CH_2_Cl_2_**, **4·3CH_2_Cl_2_**, **5**, **6**, and **7·2OCMe_2_**, respectively.

There are also several weak interactions between dirhodium units and tetraphenylphosphonium cations and solvent molecules in the structures of **3·2CH_2_Cl_2_**, **4·3CH_2_Cl_2_**, **5**, **6**, and **7·2OCMe_2_**. Each dirhodium unit of **3·2CH_2_Cl_2_** is connected to four tetraphenylphosphonium cations and two dichloromethane molecules through CH⋯O contacts. Moreover, an additional CH⋯N contact is established with another tetraphenylphosphonium cation (Appendix A). Similarly, each dirhodium unit of **4·3CH_2_Cl_2_** is connected to three tetraphenylphosphonium cations and three dichloromethane molecules through CH⋯O and CH⋯N contacts (Appendix A). Each dirhodium unit of **5** is connected to four tetraphenylphosphonium cations through CH⋯O contacts (Appendix A). Additionally, CH⋯π interactions are established with two neighboring tetraphenylphosphonium cations (Appendix A). The main cation-anion interactions in the molecular complex **6** are two CH⋯N contacts between the dicyanidoaurate(I) ligands in the axial positions of the paddlewheel units and two tetraphenylphosphonium cations (Appendix A). In the structure of **7·2OCMe_2_** each dirhodium unit is connected to two acetone molecules through CH⋯O contacts and two tetraphenylphosphonium cations through CH⋯O and CH⋯N contacts (Appendix A).

The most remarkable interactions between tetraphenylphosphonium cations are pairs of weak CH⋯π interactions established between couples of cations in the structure of **3·2CH_2_Cl_2_** (3.127 Å) and **5** (3.146 Å) (Appendix A).

## 4. Conclusions

Complexes (PPh_4_)*_n_*[Rh_2_(μ–O_2_CR)_4_Au(CN)_2_]*_n_* (R = Me (**3**), CH_2_OMe (**4**), CH_2_OEt (**5**)) show crystal structures formed by anionic wavy chains of [Rh_2_(μ–O_2_CR)_4_Au(CN)_2_]^−^ repetitive units in which the dicyanidoaurate(I) group acts as bridging ligand between the paddlewheel dirhodium(II) units. The formation of discrete anionic units, instead of anionic polymeric chains, in the complexes (PPh_4_)_2_{Rh_2_(μ–O_2_CR)_4_[Au(CN)_2_]_2_} (R = CMe_3_ (**6**), C_6_H_4_–*p*–CMe_3_ (**7**)) has been attributed to the increase of the solubility in acetone due to the branched equatorial trimethylacetate and *tert*–butylbenzoate ligands. Significant intermolecular Au⋯Au interactions and, therefore, luminescent properties, are prevented due to the presence of the bulky tetraphenylphosphonium counterions, which are also involved in several CH⋯O and CH⋯N intermolecular contacts. The many similarities found in the crystal structures of **3·2CH_2_Cl_2_**, **5**, and **6** with their silver analogous indicate that silver and gold atoms do not play a substantial role in the crystal structures of this type of complexes when the same crystallization conditions are used. The structural description of these complexes contribute to increase the family of polymers of dirhodium carboxylates with [Au(CN)_2_]^−^. This knowledge could be useful for the design of future polymers with potential applications in several areas as catalysis or bioinorganic chemistry.

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
