# Peer review of "Heteronuclear Dirhodium-Gold Anionic Complexes: Polymeric Chains and Discrete Units"

_polymers, 2020, doi:10.3390/polym12091868_

Round 1

Reviewer 1 Report

In this manuscript, the authors describe preparation of novel Rh(II) carboxylates and Au/cyanide coordination polymers based thereupon. Interestingly, although there are hundreds of complexes with Rh2(carboxylate)4 core (including polymers as well), there were only two structures of polymers with [Au(CN)2] bridge (the authors cite this work as reference 54).

I believe that such complexes may be of interest, for example, as presursors for heterometallic catalysts, so, considering the novelty of all mentioned compounds, the work is certainly suitable for publication. All experiments seem to be conducted very well, so there are no problems with experimental part.

I recommend publication of this work virtually as it is; the only recommendation is related to references. It is not obligatory, but maybe the authors could consider the following recent papers on Rh(II) carboxylates for mentioning:

1) 10.3390/molecules24030447

2) 10.1039/C8SC05733H

3) 10.1002/chem.201805833

4) 10.1002/cplu.201800513

5) 10.1016/j.ica.2012.07.028

Reviewer 2 Report

The authors provide crystal structures of dirhodium-gold anionic polymer chains and discrete complexes with two gold centers at the end. The crystal growth and refined structures are well described. The paper, however, is missing any further property description of the reported complexes. Is there a conjugation or an interaction between the gold and rhodium centers? No difference in the solid state arrangement between gold and silver (used before) could be found. Although mentioned in the introduction the physicochemical properties (magnetism or luminescence) should be interesting but they are not included hear. A mere description of the X-ray structures is not very helpful to gain interest into these complexes. Why gold was now used instead of silver used earlier, what is promising about polymeric chains 3,4 compared to the discrete 1:2 dirhodium gold complexes 6,7.

Reviewer 3 Report

This paper reported the synthesis and characterization of heteronuclear Dirhodium-Gold Anionic Complexes. This paper just presented the methods and data of these products, lacking of analysis. 

  1. Please provide the FTIR spectra as the author mentioned in part 2.2;
  2. The author mixed the Part 2 and 3, a lot of characterization were put in the materials and methods part instead of the results part;
  3. the whole paper should be reorganized for better understading. 

Round 2

Reviewer 3 Report

accept